# Is Early Oral Antimicrobial Switch Useful for Less Critically Ill Adults with Community-Onset Bacteraemia in Emergency Departments?

**DOI:** 10.3390/antibiotics9110807

**Published:** 2020-11-13

**Authors:** Ching-Chi Lee, Po-Lin Chen, Chih-Chia Hsieh, Chao-Yung Yang, Chih-Hao Lin, Wen-Chien Ko

**Affiliations:** 1Clinical Medical Research Center, National Cheng Kung University Hospital, College of Medicine, National Cheng Kung University, Tainan 70403, Taiwan; chichingbm85@gmail.com; 2Department of Internal Medicine, National Cheng Kung University Hospital, College of Medicine, National Cheng Kung University, Tainan 70403, Taiwan; cplin@mail.ncku.edu.tw; 3Department of Emergency Medicine, National Cheng Kung University Hospital, College of Medicine, National Cheng Kung University, Tainan 70403, Taiwan; hsiehchihchia@gmail.com (C.-C.H.); chao.youg@gmail.com (C.-Y.Y.); 4Department of Medicine, National Cheng Kung University Medical College, Tainan 70101, Taiwan

**Keywords:** early switch, emergency department, community, bloodstream infections

## Abstract

To compare prognoses and adverse events between bacteraemic patients in the emergency department (ED) who received an early antimicrobial IV-to-PO switch and those treated with late or no IV-to-PO switch, an 8-year multicentre cohort consisting of adults with community-onset bacteraemia was conducted. The clinical characteristics and outcomes were compared in matched cohorts by the closest propensity score calculated based on the independent determinants of 30-day mortality identified by the multivariate regression model. Of the 6664 hospitalised patients who received no or late IV-to-PO switch, 2410 were appropriately matched with 482 patients treated with early IV-to-PO switch and discharged from the ED. There were no significant differences between the two matched groups in their baseline characteristics, including the patient demographics, severity and types of comorbidities, severity and sources of bacteraemia, and the 15- and 30-day mortality rates. Notably, in addition to the shorter lengths of intravenous antimicrobial administration and hospital stay, less phlebitis and lower antimicrobial costs were observed in patients who received an early IV-to-PO switch. Similarity was observed in the clinical failure rates between the two groups. Furthermore, the inappropriate administration of empirical antibiotics and inadequate source control were identified as the only independent determinants of the post-switch 30-day crude mortality in patients who received an early IV-to-PO switch. In conclusion, for less critically ill adults with community-onset bacteraemia who received appropriate empirical antimicrobial therapy and adequate source control, an early IV-to-PO switch might be safe and cost-effective after a short course of intravenous antimicrobial therapy.

## 1. Introduction

Community-onset bacteraemia is a common disease with an annual incidence of 0.82% [1], and these systemic infections are associated with a high morbidity and mortality, incurring significant healthcare costs [2]. Particularly, in critically ill patients the rapid administration of appropriate antimicrobials is a key component and crucial therapeutic strategy for improving outcomes in patients with bloodstream infections [3,4].

For serious infections, most clinicians initially consider intravenous (IV) administration as the preferred route of antimicrobial administration. Recently, to reduce the complications arising from intravascular catheter use, excessive labour burden on nursing staff, and administration costs, numerous studies involving a vast array of infectious and septic diseases (e.g., lower respiratory tract infections [5], Gram-negative septicaemia [6,7], bloodstream infections [8,9,10], skin structure infections [11], and hospitalised patients [12,13]) have advocated for an optimal antimicrobial strategy, referred to as an “early oral switch” or “sequential antibiotic therapy”, consisting of a relatively short IV therapy followed by oral (PO) antimicrobial administration for the remainder of the treatment course. However, in the literature, there seems to be no general consensus regarding the optimal timing of the antimicrobial IV-to-PO switch because the period of IV administration varies widely (2–10 days) in previously established studies [7,14,15].

Numerous studies have reported that emergency department (ED) overcrowding is a problematic concern in many areas [16,17]; it threatens patient safety and public health because it is associated with delays in clinical diagnoses and appropriate management [18]. In principle, reducing patient demand and increasing ED capacity are essential for diminishing this overcrowding [19]. As bacteraemic patients respond clinically to empirical antibiotic therapy (EAT) and prompt management during the first 24 h after the sampling of blood cultures, physicians may be informed about the growth of blood cultures. Under the pressure of ED overcrowding and sepsis-related fatality, physicians face a stressful dilemma; a decision must subsequently be made as to whether to permit discharge or approve admission through the ED. Accordingly, it is understandable that ED physicians want to address the question, “is the strategy of early IV-to-PO switch effective for bacteraemic patients in the ED?". If this strategy is safe and efficacious, it is likely that an early switch from IV therapy to oral antibiotics is suitable to increase the ED capacity and thereby reduce ED overcrowding. Therefore, focusing on adults in the ED with community-onset bacteraemia, we compared the therapeutic efficacy and safety of early IV-to-PO switch and conventional IV therapy in matched patients using propensity scoring (PS).

## 2. Methods

### 2.1. Study Design and Setting

An eight-year, multicentre cohort study was retrospectively conducted in the EDs of three hospitals, a university-affiliated medical centre with 1200 beds, and two teaching hospitals with 460 and 380 beds in southern Taiwan during the period between January 2011 and December 2018. Adults (aged ≥18 years) with community-onset bacteraemia were included. The study was reported in accordance with the report format recommended by STROBE (Strengthening the Reporting of Observational Studies in Epidemiology) [20]. The study was approved by the institutional review board of each participating hospital (SLH 9919-108-006, SLH 9919-108-009, and B-ER-109-144), and the waiver of the requirement for informed consent was obtained under the regulations of each hospital.

### 2.2. Selection of Participants

During the study period, patients with blood cultures sampled in the ED were screened for clinically significant bacteraemia using a computer database. Focusing on all medical and surgical encounters with blood culture growth, clinical information was retrieved by reviewing medical charts. Adult patients were excluded if they had contaminated blood cultures or if they were transferred from other hospitals. Among patients with multiple bacteraemic episodes, only the first episode was taken into account. Of the remaining patients within the study endpoint, those with evidence of long courses of IV antimicrobial therapy (i.e., central nervous system infections or infective endocarditis), those that died in the ED, those with uncertain fatality within 30 days after arrival in the ED, and those with incomplete clinical information (such as uncertain bacteraemic foci) were excluded. Of the eligible patients, those who remained on IV antimicrobial therapy after hospitalisation through the ED were regarded as the group of late/no IV-to-PO switch, whereas those that were switched to oral antibiotics from initial IV antimicrobial administration in the ED and then directly discharged from the ED and accepted regular outpatient clinic visits were categorised as the group of early IV-to-PO switch.

### 2.3. Methods of Measurement

By reviewing the medical records of all eligible patients, two of the authors collected demographics and clinical characteristics in a predetermined case form. The information collected included patient age; vital signs; types and severity of comorbidities; laboratory data; the duration, type, and cost of antimicrobial agents; microbiological results; imaging studies; the source of bacteraemia; the dates and types of radiologic or surgical interventions; bacteraemia sources; bacteraemia and comorbid severity (by the Mortality in Emergency Department Sepsis (MEDS) score) at onset; and the length of hospitalisation. For the clinical information assessed at bacteraemia onset, the worst record for each variable during the ED stay was captured. For each patient, two authors inspected the medical records together for consensus, and any discrepancies between authors in capturing medical information were resolved through discussion with another author.

### 2.4. Outcome Measures

The primary endpoint was crude mortality within 30 days of bacteraemia onset (i.e., ED arrival), and the secondary endpoint was post-switch mortality within 30 days after the discontinuation of IV antibiotic administration (i.e., after IV-to-PO switch). Primary exposure was the duration of IV antimicrobial administration, dichotomised to extremely short-course (approximately 1 day) and conventionally long-course (6 days) therapy. Therapeutic efficacies (i.e., the clinical failure of IV-to-PO switch) within 30 days after IV-to-PO switch included post-switch crude mortality, ED revisits, re-hospitalisation, or the re-initiation of IV antimicrobial treatment.

### 2.5. Definitions

Bacteraemia was defined as bacterial growth in blood cultures drawn from central or peripheral venepuncture, after the exclusion of contaminant sampling in accordance with previously established criteria [21]. The term “community-onset bacteraemia” is generally understood to mean that the bacteraemia was acquired in the community, and includes health care- and community-associated bacteraemia [3,4]. The isolation of more than one microbial species in one bacteraemic episode was regarded as polymicrobial bacteraemia.

As previously defined [3,4], antimicrobial therapy was appropriate if all of the following criteria were fulfilled: (i) the route and dosage of antimicrobials were administered as recommended in the *2020 Sanford Guide* [22], and (ii) the causative microorganisms were susceptible to the administered antimicrobials in vitro, based on the contemporary breakpoints of the Clinical and Laboratory Standards Institute (CLSI) criteria issued in 2020 [23]. The time-to-appropriate IV antibiotics (TtAa) was defined as the period between ED arrival (i.e., ED triage) and the initial IV administration of appropriate antimicrobials, and empirical administration was considered appropriate if the TtAa was ≤24 h [3,24].

In agreement with previous concepts [25,26], a patient was assigned to one of the primary sources of bacteraemia based on the clinical diagnosis and/or isolation of pathogens. Primary bacteraemia was classified as a bacteraemic source that could not be assigned to a specific site. As previously described [27], complicated bacteraemia was defined by the determining whether a bacteraemic source was amenable to source control, such as the drainage of an abscess or obstructive tract, the removal of a potentially infected device, the debridement of infected necrotic tissue, or the definitive control of a source of ongoing microbial contamination. In the literature, there seems to be no general definition of the appropriateness of source control for complicated bacteraemia; consistent with definitions in previously established studies [28,29], the sources of bacteraemia and appropriateness of specific percutaneous or surgical source control were determined by physicians trained in infectious diseases.

The bacteraemia severity was graded based on MEDS scoring, a scoring system based on the worst physiological variables in the ED and ranging from 0 to 27 points depending on whether variables were present or absent [30]. The prediction rules of MEDS scoring stratified patients into five mortality risk groups: very low, 0–4 points; low, 5–7; moderate, 8–12; high, 13–15; and very high, >15 [30]. Accordingly, patients with a MEDS score of <8 were grouped as less critically ill patients. Comorbidities were categorised based on previous criteria [31] and malignancies included haematological malignancies and solid tumours. Comorbid severity was defined by McCabe and assessed based on a delineated classification [32].

### 2.6. Microbiological Methods

Blood cultures were incubated in a BACTEC 9240 instrument (Becton Dickinson Diagnostic Systems, Sparks, MD, USA) for 5 days at 35 °C. Bacteraemic aerobic isolates from the study period were prospectively collected. Bacterial species were identified by the Vitek-two system (bioMérieux, Durham, NC, USA), and antimicrobial susceptibilities were determined by the disk diffusion method, based on the contemporary CLSI standard [23]. For Gram-negative microorganisms, the tested drugs included ertapenem, cefepime, cefotaxime, cefuroxime, cefazolin, ampicillin/sulbactam, and levofloxacin. For streptococci, penicillin, cefotaxime, and levofloxacin were tested. Cefoxitin and ampicillin were tested for staphylococci and enterococci, respectively. If a patient was treated empirically with any other antimicrobial agents, their susceptibilities were measured.

### 2.7. Statistical Analyses

The Statistical Package for the Social Sciences for Windows (Version 23.0; Chicago, IL, USA) was used for the statistical analyses. The Fisher’s exact or Pearson’s chi-square test was used to examine categorical variables and an independent *t* or the Mann–Whitney test was used for continuous variables. To recognise the independent determinants, regardless of the overall or matched cohorts, the variable of 30-day mortality in the univariate analysis with a *p* value of <0.05 was conducted in the logistic regression model (stepwise and backward). A 2-sided *p* value of <0.05 was considered significant.

A PS-matched analysis was performed to control for confounding variables in the choice of which antimicrobial administration route to continue (with early or no IV-to-PO switch) made by ED clinicians. For the overall cohort. the PS was calculated based on the independent determinant of 30-day crude mortality identified by the logistic regression model. Patients who received the early IV-to-PO switch were matched at a ratio of 1:5 with those with late or no IV-to-PO switch by individual PSs. Matching based on the closest scores was conducted manually based on a tolerance interval approach in accordance with a matching tolerance of a PS difference of 0.2 [33].

## 3. Results

### 3.1. Characteristics of Study Subjects

The total of 7146 adults with community-onset bacteraemia categorised by early (482 patients, 6.7%) or late/no IV-to-PO switch (6664, 93.3%) were included based on the inclusion and exclusion criteria (Figure 1). Of the entire cohort, the median (interquartile range, IQR) age was 70 (57–80) years old and 3670 (51.4%) were males. The median (IQR) TtAa was 2 (1–11) hours, and patients receiving inappropriate EAT only accounted for 19.2% (1372 patients) of the total. Patients with complicated bacteraemia accounted for 21.1% (1505 patients), and the majority of these patients (1265, 84.1%) received adequate source control during IV antimicrobial therapy. The 3-day, 15-day, and 30-day crude mortality after the onset of bacteraemia (i.e., ED arrival) was 3.1% (220 patients), 8.1% (576), and 11.8% (846), respectively.

Out of 682 episodes of polymicrobial bacteraemia, a total of 8032 causative pathogens were collected. The major pathogens included *E. coli* (2978, 37.1%), *Klebsiella* species (1259, 15.7%), *Streptococcus* species (1028, 12.8%), *Staphylococcus aureus* (825, 10.3%), *Pseudomonas* species (246, 3.1%), *Enterococcus* species (238, 3.0%), *Enterobacter* species (187, 2.3%), *Proteus* species (182, 2.3%), *Salmonella* species (127, 1.6%), and *Aeromonas* species (99, 1.2%). Overall, cefazolin was active against 57.6% (1715/2978), 81.7% (1028/1259), and 65.4 (104/159) of *E. coli*, *Klebsiella* species, and *Proteus mirabilis* (EKP)*,* respectively. Of 4396 EKP isolates, the susceptibilities to cefotaxime, cefepime, ertapenem, and levofloxacin were 80.3% (3528 isolates), 92.7% (4077), 99.3% (4366), and 80.0% (3515), respectively. Penicillin- and levofloxacin-susceptible isolates accounted for 95.7% (984) and 97.8% (1005) of 1028 streptococci, respectively. Ampicillin-susceptible enterococci accounted for 84.5% (201/238) of enterococci and MRSA isolates accounted for 39.2% (323/825) of *S. aureus*.

### 3.2. Baseline Characteristics and Outcomes between the Early and Late/No IV-to-PO Switch Groups

As shown in Table 1, the univariate analysis demonstrated vast dissimilarities between the early and late/no switch groups in the baseline characteristics at bacteraemia onset and outcomes for the overall cohort. The elderly; nursing-home residents; high-risk patients (by MEDS score); patients with fatal comorbidities (by McCabe classification); patients with comorbidities of hypertension, diabetes mellitus, malignancies, neurological diseases, or urological diseases; and patients with bacteraemic pneumonia, bacteraemia due to biliary tract infections or liver abscess, and complicated bacteraemia were frequently observed in the late/no switch group, compared to the early switch group. However, a smaller proportion of patients with bacteraemia due to urinary tract or skin and soft-tissue infections were observed in the late/no switch group. Moreover, longer periods of IV antimicrobial administration, total antimicrobial administration, and total hospitalisation as well as shorter lengths of ED stay and higher 15- and 30-day mortality rates were observed in the late/no switch group compared to the early switch group. Notably, the proportion of patients receiving inappropriate EAT and inadequate source control during IV antimicrobial administration was similar between the two groups.

### 3.3. Predictors of 30-Day Crude Mortality

Using the univariate analysis, several variables were positively associated with the 30-day crude mortality (Table 2), including being elderly, male, a nursing-home resident, or a high-risk patient at bacteraemia onset; receiving inappropriate EAT or inadequate source control; having polymicrobial bacteraemia, bacteraemic pneumonia, or bacteraemia due to intraabdominal infections; and having comorbidities including malignancies, neurological diseases, chronic kidney diseases, or liver cirrhosis. By contrast, bacteraemia due to urinary tract infections, biliary tract infections, or liver abscess and comorbid hypertension or diabetes mellitus were associated with favourable prognoses.

Of numerous predictors of 30-day crude mortality identified by the univariate analysis (Table 2), 12 independent predictors were identified in the multivariate regression model, including being elderly, a nursing-home resident, or a high risk patient; receiving inadequate source control or inappropriate EAT; having polymicrobial bacteraemia, bacteraemic pneumonia, or bacteraemia due to intraabdominal infections, urinary tract infections, or liver abscess; and having malignancies or liver cirrhosis as comorbidities. Of note, in addition to severe bacteraemic episodes, inappropriate EAT and inadequate source control were the most important determinates linked to unfavourable prognoses.

### 3.4. Clinical Characteristics and Outcomes of the PS-Matched Cohort

Of 6664 patients with late/no IV-to-PO switch, 2410 were matched with 482 in the early switch group with the closest PS based on 12 independent predictors of 30-day mortality. After PS-matching, no significant differences in age, gender, nursing-home residency, major comorbidities, comorbidity severity, bacteraemia severity at onset, or major sources of bacteraemia were observed between the two matched groups (Table 1). Although fewer patients with complicated bacteraemia were observed in the group of early IV-to-PO switch, the proportion of patients receiving inappropriate EAT or inadequate source control was similar between the two matched groups.

Consequently, shorter durations of IV antimicrobial administration and hospital stay were observed in the group of early IV-to-PO switch, but the total period (i.e., oral plus IV-form) of antimicrobial administration and the 15- and 30-day mortality rates were similar in the two matched groups (Table 1). More importantly, the clinical failure rate within 30 days of IV-to-PO switch in terms of post-switch crude mortality, ED revisits, re-hospitalisation, and the re-initiation of IV antimicrobial administration was similar in the two matched groups; however, more phlebitis, less gastrointestinal intolerance, and higher antimicrobial costs remained significant in the late/no switch group (Table 3).

### 3.5. Predictors of Post-Switch 30-Day Crude Mortality in Early Switch Patients

Focusing on the 482 patients who received early IV-to-PO switch (Table 4), eight predictors of crude mortality within 30 days after IV-to-PO switch were identified by the univariate analysis: polymicrobial bacteraemia, inappropriate EAT, inadequate source control, moderate- and high-risk patient status at ED arrival, neurological or chronic kidney disease as a comorbidity, and bacteraemia due to pneumonia or urinary tract infections. However, inappropriate EAT and inadequate source control during IV antimicrobial therapy were the only two independent determinants of mortality under the multivariate regression model.

## 4. Discussion

Generally, clinicians agree in principle that the optimal way to achieve a rapid onset of drug action is to create an instant therapeutic effect by administering via the IV route. However, over the past 20 years economic pressure has influenced medical culture to shift from conventional IV therapy for the entire therapeutic course to an early switch to oral administration or home administration of IV antibiotics. Therefore, for various types of infection, numerous randomised clinical trials have compared IV therapy over the entire treatment course with short-course of IV therapy followed by oral therapy to establish a consensus for medically complex populations [5,6,7,11,34]. Although numerous review articles have concluded that the evidence supports the efficacy of this switch strategy [14,15,35], little is known about the consequences of introducing this strategy in ED patients with bloodstream infections. Interestingly, the incorporation of an early switch strategy into the antimicrobial stewardship programme for bacteraemic patients in the ED was first adapted in this study after an extremely short course (approximately 1 day) of appropriate antimicrobial IV administration. More importantly, before adapting early IV-to-PO switch for bacteraemic patients in the ED, the appropriate administration of EAT and adequate source control (if the patient has complicated bacteraemia) was essential for achieving an improved prognosis.

Using a retrospective design to collect information on a long-term bacteraemia cohort was a reasonable necessity in the present study to ensure patient safety when a short course of IV antimicrobial therapy was applied. Furthermore, the decisions made by ED clinicians to admit or discharge patients through the ED can be stressful and were influenced by many clinical situations herein, including the severity of comorbidities, infectious sources, bacteraemic severity, the degree of ED crowding, and the availability of hospital beds. For example, it is probable that the shorter ED stays of patients who received early IV-to-PO switch might be associated with ED overcrowding or a low availability of hospital beds. For this type of observational data, PS analysis is generally a more favourable approach with which to assess causal effects using clinical information compared to traditional regression analyses [10]. Despite the unmeasured confounding factors in our dataset, such as the bioavailability of oral antibiotics, the manpower of ED teams, and the degree of ED overcrowding, PS-matching was attempted to overcome the differences in baseline characteristics and to minimise potential biases between the two groups. As a result, PS-matching was performed and resulted in two closely matched groups, with the main difference in our cohort being the IV antibiotic duration.

Several severity scoring systems, such as the Pitt bacteraemia score [3]; the Sepsis-related Organ Failure Assessment (SOFA) score [36]; the Simplified Acute Physiology Score (SAPS) [37]; and the Acute Physiology, Age, Chronic Health Evaluation (APACHE) score [38], have been proposed to predict the outcomes of septic patients. However, most of them formulate their corresponding scores based on parameters that are unavailable to ED physicians. Because MEDS scoring has been validated for predicting the prognoses of patients with bloodstream infections [39] and because this scoring algorithm consists of parameters that are available during an ED stay, our study utilised the MEDS score as an indicator of bacteraemia severity. In addition, to provide a convenient reference for ED physicians, the worst records for physiological variables and parameters were captured during the ED stay. Based on the MEDS score, ED clinicians can rapidly identify less critically ill patients and then engage the IV-to-PO switch strategy to reduce ED overcrowding if their patient had already appropriately received an extremely short-course therapy of IV antimicrobials and source control (in cases of complicated bacteraemia).

These results need to be interpreted with caution for the following reasons. First, patients who had previously experienced numerous infections requiring long-term IV antimicrobial therapy as the standard therapy, such as infective endocarditis and central nervous system infections, and those who died during their ED stay were excluded from our cohort. Moreover, PS-matching was used to overcome differences in the baseline characteristics between the early and late/no IV-to-PO switch groups. During the matching process, a substantial portion of the critically ill patients was excluded because of vast dissimilarities in the baseline characteristics between the two original patient groups. Therefore, our findings should be limited to bacteraemic patients who initially present with less critical illness. Second, it was understandable that a wide range of antimicrobial classes had been administered in our cohort for different clinical conditions and various types of bacteraemia. Similar to several other studies dealing with early IV-to-PO switch [5,12,13], assessing the impact of different antimicrobial regimens on clinical outcomes would be difficult and thus was not discussed herein. Finally, each participating hospital was localised to south Taiwan; a lack of independent validation is one of the study limitations, and our results may not be extended to other populations with varied comorbidity types and causative microorganisms (particularly for antibiotic-resistant pathogens). However, due to the strengths of the large multicentre cohort, an observational multicentre study designed to prospectively solve the problem of the ED overcrowding should be performed in the future based on our findings.

## 5. Conclusions

For the less critically ill adults, indicted by a MEDS score of <8, with community-onset bacteraemia who received appropriate EAT and adequate source control (in cases of complicated bacteraemia), an early switch to oral agents after an extremely short course (approximate 1 day) of IV therapy might be effective and less costly. Accordingly, our findings should be incorporated in an antibiotic stewardship programme to help relieve ED overcrowding.

## Figures and Tables

**Figure 1 antibiotics-09-00807-f001:**
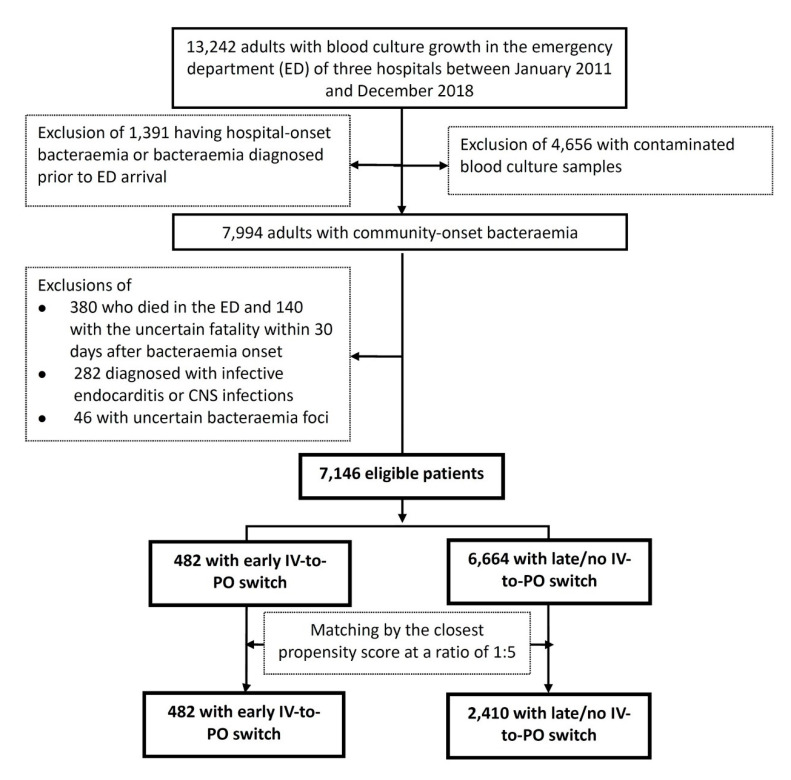
Flowchart of patient selections. CNS = central nerve system; ED = emergency department; MEDS = Mortality in Emergency Department Sepsis.

**Table 1 antibiotics-09-00807-t001:** Clinical characteristics and mortality rates of the early and late/no IV-to-PO switch patients in the overall and matched cohorts *.

Clinical Variables	Patients Number (%)					
	Overall Cohort			Matched Cohort		
	Early Switch N = 482	Late/No Switch N = 6664	*p* Value	Early Switch N = 482	Late/No Switch N = 2410	*p* Value
Gender, male	236 (49.0)	3434 (51.5)	0.28	236 (49.0)	1174 (48.7)	0.92
Elderly, ≥65 years	229 (47.5)	4086 (61.3)	<0.001	229 (47.5)	1198 (49.7)	0.38
Nursing-home residents	8 (1.7)	405 (6.1)	<0.001	8 (1.7)	63 (2.6)	0.22
Inadequate source control during IV antimicrobial administration	14 (2.9)	226 (3.4)	0.57	14 (2.9)	49 (2.0)	0.23
Inappropriate empirical antimicrobial therapy	82 (17.0)	1290 (19.4)	0.21	82 (17.0)	414 (17.2)	0.93
Periods of antimicrobial administration, days, median (IQR)						
**IV (before IV-to-PO switch)**	**0.9 (0.3–2.0)**	**6 (6–16)**	**<0.001**	**0.9 (0.3–2.0)**	**9 (6–14)**	**<0.001**
Total	13.0 (10.5–14.3)	10 (13–18)	<0.001	13.0 (10.5–14.3)	13 (10–16)	0.12
Lengths of stay, median (IQR)						
**Emergency department, hours**	**24.0 (7.3–52.8)**	**16.0 (5.9–27.0)**	**<0.001**	**24.0 (7.3–52.8)**	**15.7 (6.0–16.2)**	**<0.001**
**Total hospitalisation, days**	**1.0 (0.3–2.2)**	**11 (7–19)**	**<0.001**	**1.0 (0.3–2.2)**	**7 (10–16)**	**<0.001**
Polymicrobial bacteraemia	36 (7.5)	646 (9.7)	0.11	36 (7.5)	170 (7.1)	0.75
**Complicated bacteraemia**	**36 (7.5)**	**1469 (22.0)**	**<0.001**	**36 (7.5)**	**490 (20.3)**	**<0.001**
MEDS scores at bacteraemia onset			<0.001			
High risk (≥13)	30 (6.2)	1596 (23.9)		30 (6.2)	155 (6.2)	0.87
Low risk (<8) *	358 (74.3)	3062 (45.9)		358 (74.3)	1718 (71.3)	0.18
Comorbidity severity (McCabe classification)			<0.001			0.06
Ultimately and rapidly fatal	88 (18.3)	1766 (26.5)		88 (18.3)	357 (14.8)	
Nonfatal	394 (81.7)	4898 (73.5)		394 (81.7)		
Major comorbidities						
Hypertension	201 (41.7)	3290 (49.4)	0.001	201 (41.7)	1121 (46.5)	0.05
Diabetes mellitus	129 (26.8)	2632 (39.5)	<0.001	129 (26.8)	751 (31.2)	0.07
Malignancies	108 (22.4)	2107 (31.6)	<0.001	108 (22.4)	553 (22.9)	0.80
Neurological diseases	78 (16.2)	1607 (24.1)	<0.001	78 (16.2)	440 (18.3)	0.28
Chronic kidney diseases	92 (19.1)	1282 (19.2)	0.94	92 (19.1)	413 (17.1)	0.30
Liver cirrhosis	48 (10.0)	854 (12.8)	0.07	48 (10.0)	201 (8.3)	0.25
Coronary artery diseases	43 (8.9)	662 (9.9)	0.47	43 (9.1)	211 (8.8)	0.91
Urological disorders	30 (6.2)	520 (9.3)	0.02	30 (6.2)	210 (8.7)	0.07
Major sources of bacteraemia						
Urinary tract	192 (39.8)	2267 (34.0)	0.009	190 (39.4)	921 (38.2)	0.62
Skin and soft-tissue infections	72 (14.9)	784 (11.8)	0.04	72 (14.9)	341 (14.1)	0.65
Intraabdominal	58 (12.0)	810 (12.2)	0.94	58 (12.0)	285 (11.8)	0.90
Pneumonia	54 (11.2)	985 (14.8)	0.03	54 (11.2)	233 (9.7)	0.30
Biliary tract	25 (5.2)	614 (9.2)	0.003	25 (5.2)	185 (7.7)	0.06
Liver abscess	4 (0.8)	270 (4.1)	<0.001	4 (0.8)	32 (1.3)	0.37
Crude mortality rates after bacteraemia onset						
15-day	12 (2.5)	564 (8.5)	<0.001	12 (2.5)	60 (2.5)	1.00
30-day	16 (3.3)	830 (12.5)	<0.001	16 (3.3)	111 (4.6)	0.21

IQR = interquartile range; IV = intravenous; MEDS = Mortality in Emergency Department Sepsis. Data are given as number (percent), unless otherwise specified. * Patients having a MEDS score of <8 were grouped as less critically ill patients. Boldface indicates statistical significance under the backward multivariate regression—i.e., a *p* value of <0.05.

**Table 2 antibiotics-09-00807-t002:** Risk factors of 30-day crude mortality in a total of 7146 patients.

Variables	Patient Number (%)		Univariate Analysis		Multivariate Analysis	
	Death, N = 846	Survival, N = 6300	OR (95% CI)	*p* Value	Adjusted OR (95% CI)	*p* Value
Gender, male	514 (60.8)	3156 (50.1)	1.54 (1.33–1.79)	<0.001	NS	NS
**Elderly**	**551 (65.1)**	**3764 (59.7)**	**1.25 (1.08–1.46)**	**0.003**	**1.31 (1.09–1.57)**	**0.004**
**Nursing-home residents**	**108 (12.8)**	**306 (4.9)**	**2.87 (2.27–3.62)**	**<0.001**	**1.42 (1.07–1.89)**	**0.02**
**High risk patients (MEDS score ≥ 13) at bacteraemia onset**	**610 (72.1)**	**1016 (16.1)**	**13.44 (10.40–15.85)**	**<0.001**	**9.60 (7.93–11.63)**	**<0.001**
**Inadequate source control during IV antimicrobial administration**	**56 (6.6)**	**184 (2.9)**	**2.36 (1.73–3.21)**	**<0.001**	**3.72 (2.54–5.44)**	**<0.001**
**Inappropriate empirical antibiotic therapy**	**212 (25.1)**	**1160 (18.4)**	**1.48 (1.25–1.75)**	**<0.001**	**1.97 (1.60–2.41**	**<0.001**
**Polymicrobial bacteraemia**	**142 (16.8)**	**540 (8.6)**	**2.15 (1.76–2.63)**	**<0.001**	**1.62 (1.27–2.07)**	**<0.001**
Bacteraemia sources						
**Pneumonia**	**341 (40.3)**	**698 (11.1)**	**5.42 (4.63–6.35)**	**<0.001**	**2.09 (1.67–2.63)**	**<0.001**
**Intraabdominal**	**134 (15.8)**	**734 (11.7)**	**1.43 (1.17–1.74)**	**<0.001**	**1.46 (1.11–1.91)**	**0.006**
**Urinary tract**	**122 (14.4)**	**2337 (37.1)**	**0.29 (0.23–0.35)**	**<0.001**	**0.50 (0.39–0.64)**	**<0.001**
Biliary tract	46 (5.4)	593 (9.4)	0.55 (0.41–0.75)	<0.001	0.66 (0.46 -1.01)	0.06
**Liver abscess**	**14 (1.7)**	**260 (4.1)**	**0.39 (0.23–0.67)**	**<0.001**	**0.40 (0.22–0.71)**	**0.002**
Comorbidities						
**Malignancies**	**441 (52.1)**	**1774 (28.2)**	**2.78 (2.40–3.21)**	**<0.001**	**2.02 (1.70–2.39)**	**<0.001**
Hypertension	350 (41.4)	3141 (49.9)	0.71 (0.61–0.82)	<0.001	NS	NS
Diabetes mellitus	294 (34.8)	2467 (39.2)	0.83 (0.71–0.96)	0.01	NS	NS
**Liver cirrhosis**	**188 (22.2)**	**714 (11.3)**	**2.24 (1.87–2.68)**	**<0.001**	**1.32 (1.06–1.65)**	**0.01**
Neurological diseases	238 (28.1)	1447 (23.0)	1.31 (1.12–1.54)	0.001	NS	NS
Chronic kidney diseases	184 (21.7)	1190 (18.9)	1.19 (1.00–1.42)	0.047	NS	NS

IV = intravenous; MEDS = Mortality in Emergency Department Sepsis; NS = not significant (after processing the backward multivariate regression); OR = odds ratio. Boldface indicates statistical significance under the backward multivariate regression—i.e., a *p* value of <0.05.

**Table 3 antibiotics-09-00807-t003:** The antimicrobial cost, intolerability, and clinical failure rate of matched patients receiving early and late/no switch.

Variables	Patient Numbers (%)		*p* Values
	Early Switch N = 482	Late/No Switch N = 2410	
Antimicrobial costs (US $), median (IQR)			
**IV**	**6.1 (3.2–7.5)**	**146.7 (44.2–300.0)**	**<0.001**
**Oral**	**12.6 (10.2–19.0)**	**3.4 (2.1–4.8)**	**<0.001**
**Total**	**11.2 (8.6–36.8)**	**153.8 (48.6–302.9)**	**<0.001**
**Daily antimicrobial costs** (US $), median (IQR)	**0.9 (0.6–2.9)**	**11.8 (3.8–23.3)**	**<0.001**
Intolerability of antimicrobial therapy			
**Phlebitis**	**4 (0.8)**	**55 (2.3)**	**0.04**
**Gastrointestinal intolerance**	**12 (2.5)**	**26 (1.1)**	**0.01**
Clinical failure rate within 30 days of IV-to-PO switch			
Post-switch crude mortality	16 (3.3)	55 (2.3)	0.18
ED revisit	14 (2.9)	65 (2.7)	0.80
Re-hospitalisation	10 (2.1)	55 (2.3)	0.78
Re-initiation of IV antimicrobials	12 (2.5)	39 (1.6)	0.19

ED = emergency department; IOR = interquartile range; IV = intravenous. Boldface indicates statistical significance—i.e., a *p* value of <0.05.

**Table 4 antibiotics-09-00807-t004:** Predictors of post-switch 30-day mortality in 482 patients who received early IV-to-PO switch.

Variables	Patient Numbers (%)		Univariate Analyses		Multivariate Analyses	
	Death N = 16	Survival N = 466	OR (95% CI)	*p* Value	Adjusted OR (95% CI)	*p* Value
Polymicrobial bacteraemia	4 (25.0)	32 (6.9)	4.52 (1.38–14.82)	0.03	NS	NS
**Inappropriate empirical antibiotic therapy**	**8 (50.0)**	**74 (15.9)**	**5.30** **(1.93–14.56)**	**0.002**	**15.24** **(3.45–67.32)**	**<0.001**
**Inadequate source control during IV antimicrobial administration**	**4 (25.0)**	**10 (2.1)**	**15.20** **(4.17–55.41)**	**<0.001**	**73.15** **(9.21–581.34)**	**<0.001**
Moderate- and high-risk patients (MEDS score ≥ 8)	16 (100)	108 (23.2)	–	<0.001	NS	NS
Comorbidities						
Neurological diseases	8 (50.0)	70 (14.0)	5.66 (2.06–15.57)	0.001	3.68 (0.90–17.62)	0.07
Chronic kidney diseases	8 (50.0)	84 (18.0)	4.55 (1.66–12.45)	0.004	NS	NS
Bacteraemia sources						
Pneumonia	6 (37.5)	48 (10.3)	5.02 (1.82–15.01)	0.005	NS	NS
Urinary tract	2 (12.5)	190 (40.8)	0.21 (0.05–0.92)	0.02	NS	NS

CI = confidence interval; IV = intravenous; MEDS = Mortality in Emergency Department Sepsis; NS = not significant (after processing the backward multivariate regression); OR = odds ratio. Boldface indicates statistical significance—i.e., a *p* value of <0.05.

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
