# Peer review of "Is Early Oral Antimicrobial Switch Useful for Less Critically Ill Adults with Community-Onset Bacteraemia in Emergency Departments?"

_antibiotics, 2020, doi:10.3390/antibiotics9110807_

Round 1
Reviewer 1 Report
This is a retrospective cohort study of 7146 patients from the community presenting to the emergency room and later proving to have positive blood cultures, subsequently receiving either a full-length hospital stay with intravenous antibiotics, or a short 24 hour course of i.v. antibiotics followed by an early oral antimicrobial switch (OAS). The treatment decision was done at the physicians' clinical discretion. Patients were recruited from multiple centres in Taiwan over an eigth year period. Comparisons between the in-patient and OAS groups were done after propensity matching, and showed no differences in clinical failure rates, but a higher 30-day mortality in the early OAS group in those with inappropriate administration of empirical antibiotics and inadequate infection source control. The authors conclude that in less critically ill adults with bacteremia who received appropriate empirical antibiotics and infection source control, early oral antimicrobial switch might be safe and cost-effective.
The strengths of the paper are that the material is large and multi-center. Also, the use of statistics here is appropriate, with matching using propensity scoring, and multivariate logistic regression models to evaluate risk factors for mortality. Also, tables are clean and easy to read.
The weaknesses are several and of major importance. First, to be able to truly answer the question at hand - is an early oral switch as safe as a full course of intravenous antibiotics? - one would have to design a randomised controlled trial. Based on the present study, with its inherent selection bias in allocating patients at the clinicians' discretion to one or the other treatment arm, this reviewer thinks that this study cannot really answer the question whether an early oral switch is as safe as a full course of intravenous antibiotics. This should be reflected better in the Discussion. However, the authors should be credited for including multivariate regression analyses to find associations between risk factors and outcomes, which makes excellent use of the material given the limitations due to the mentioned selection bias.
Also, the authors limit the generalizability of the study to patients who are less critically ill. The reasons why the study is valid for only a subset of patients is not clearly stated. Also, no distinct cut-off for clinical severity scoring is presented, so that the concept of "less critically ill" remains vague.
Of minor importance:
- The process of propensity matching is unclearly described, and should be rewritten for clarity.
- The paragraph with text lines 379 to 397 should be moved from the Results to the Discussion section.
In summary, the authors should be credited for a large study with many participants, but the study design carries with it inherent biases that makes the paper unable to correctly answer the primary scientific question posed.
Author Response
- This is a retrospective cohort study of 7146 patients from the community presenting to the emergency room and later proving to have positive blood cultures, subsequently receiving either a full-length hospital stay with intravenous antibiotics, or a short 24 hour course of i.v. antibiotics followed by an early oral antimicrobial switch (OAS). The treatment decision was done at the physicians' clinical discretion. Patients were recruited from multiple centres in Taiwan over an eight year period. Comparisons between the in-patient and OAS groups were done after propensity matching, and showed no differences in clinical failure rates, but a higher 30-day mortality in the early OAS group in those with inappropriate administration of empirical antibiotics and inadequate infection source control. The authors conclude that in less critically ill adults with bacteremia who received appropriate empirical antibiotics and infection source control, early oral antimicrobial switch might be safe and cost-effective. The strengths of the paper are that the material is large and multi-center. Also, the use of statistics here is appropriate, with matching using propensity scoring, and multivariate logistic regression models to evaluate risk factors for mortality. Also, tables are clean and easy to read
Response: Many thanks for your review and substantial opinions.
.
- The weaknesses are several and of major importance. First, to be able to truly answer the question at hand - is an early oral switch as safe as a full course of intravenous antibiotics? - one would have to design a randomized controlled trial. Based on the present study, with its inherent selection bias in allocating patients at the clinicians' discretion to one or the other treatment arm, this reviewer thinks that this study cannot really answer the question whether an early oral switch is as safe as a full course of intravenous antibiotics. This should be reflected better in the Discussion. However, the authors should be credited for including multivariate regression analyses to find associations between risk factors and outcomes, which makes excellent use of the material given the limitations due to the mentioned selection bias.
Response: Thanks for your question and opinion. Although, the strategy of early IV-to-PO switch was widely adapted in numerous clinical setting (Line 65-75, Page2). The safety of patient receiving the extremely short course of IV antimicrobial therapy following by oral antimicrobial therapy remained limited. For this concern, the study design for establishing the retrospective cohort was safe (Line 406-408, Page 16). On the other hand, as the point indicated by Dr Joe Amoah and his colleagues (Reference No. 10), "Propensity score analysis is overall a more favorable approach than traditional regression analysis when estimating causal effects using the observational cohort. As with all analytic methods using observational data, residual confounding will remain; only variables that are measured can be accounted for. Moreover, propensity-score analysis does not compensate for poor study design or questionable data accuracy", the adaption of propensity score matching to overcome the confounding factor in the groups of early and no/late IV-to-PO switch was reasonable herein. Please refer to line 409-424 (Page 16) in the revised manuscript.
In addition to PS matching, focusing on the overall cohort and patients who received early IV-to-PO switch, multivariate analyses were respectively using to identify the risk factors of mortality. Please refer to the section of "2.7 statistical analyses" (Line 219-222, Page 6).
- Also, the authors limit the generalizability of the study to patients who are less critically ill. The reasons why the study is valid for only a subset of patients is not clearly stated. Also, no distinct cut-off for clinical severity scoring is presented, so that the concept of "less critically ill" remains vague.
Response: Thanks for your opinion. Through the propensity-score matching, the majority of patients initially presented with the critical illness in the group of late/no IV-to-PO switch was excluded from the further analyses (Line 446-452, Page 18). Therefore, in the matching cohort, the less critically ill patients, defined by the MEDS score of < 8 (Line 194-195, Page 6), both accounted for 74.3% and 71.3% (Table 1) in the early and late/no switch groups, respective. So, our finding was generalizable to other critically ill population. For reader's more understandings, the sentence was revised for highlight in the Table 1 footnote (Line 323), and the section of conclusion (Line 467, Page 18).
- Of minor importance:
(1) The process of propensity matching is unclearly described, and should be rewritten for clarity.
Response: Thanks for your suggestion. the process of PS-matching was revised in the section of "2.7. Statistical Analyses" (Line 224-232, Page 6-7).
(2) The paragraph with text lines 379 to 397 should be moved from the Results to the Discussion section.
Response: This paragraph was moved to the section of discussion. Please refer to line 442-464 on page 17.
(3) In summary, the authors should be credited for a large study with many participants, but the study design carries with it inherent biases that makes the paper unable to correctly answer the primary scientific question posed.
Response: Thanks for your suggestion. During the study period, the Antimicrobial Stewardship Program was not performed in the ED of each participating hospital. So, to solve the problem of the ED overcrowding, we were interested in understanding the patient safety after extremely early IV-to-PO switch using the retrospective cohort. Based our finding, the further prospective study will be conducted after incorporation of an early switch strategy into the antimicrobial stewardship programme for bacteraemic patients diagnosed in the ED. Some sentences were inserted for our further plan (Line 461-464, Page 17 and Line 471-473, Page 18).
Reviewer 2 Report
The authors have conducted a well-designed and analyzed study re: early oral antimicrobial switch for less critically ill adults with community-onset bacteremia in the ED setting. The study has a robust design and statistical analysis (i.e. a large sample size over the course of eight years, propensity score matching). It is a very well written paper - thank you for the contribution to this important infectious disease topic.
MAJOR COMMENTS:
- Consider mentioning or referring to standardized term of “IV-to-PO” since this terminology is widely adopted in antimicrobial stewardship literature? Standardization of “IV-to-PO” may appeal more to readers due to familiarity.
- Consider reviewing and including the following reference on IV-to-PO switch in bacteremia pts: Erickson, Reaghan et al. Open Forum Infect Dis. 2019 Nov 13;6(12):ofz490. doi: 10.1093/ofid/ofz490. eCollection 2019 Dec. Impact of an Antimicrobial Stewardship Bundle for Uncomplicated Gram-Negative Bacteremia.
- Were rapid diagnostics (i.e. MALDI-TOF, PCR, PNA-FISH, BC-GP) available for the study sites?
- Appreciate the cost analysis in Table 3, have authors consider direct costs (antimicrobial costs) vs. cost-effectiveness calculations (deaths averted, quality-adjusted life years gained (QALYs), incremental cost-effectiveness ratios (ICERs))? Also, consider calculations of defined daily dose (DDD) of antibiotics and/or days of therapy (DOT) since these are standard methods of quantifying antibiotic use? An extensive cost analysis could be a separate discussion or paper as a follow-up to the current manuscript.
- Early vs. late PO switch was not clearly defined. Some of the definition was covered in section 2.4 Outcome Measures re: extremely short course vs. conventionally long-course. Consider clearly defining what is early vs. late switch in the context of this study. Duration of IV antimicrobial use before switch was not shared for the study population (i.e. median, range, etc.). There was some information in section 4. Discussion where the authors shared that extremely short course was approximately 1 day of antimicrobial IV administration. It would be helpful to include more information about this earlier in the paper.
- Did each participating study site have an Antimicrobial Stewardship Program in place? Did the antimicrobial steward or pharmacists make the recommendations for the IV-to-PO switches or did the house staff make the switches on their own? It would be helpful to elaborate a bit about the resources and processes in place that allowed for the IV-to-PO switch to happen in the case other institutions would like to implement something similar.
- Would be interested in authors sharing in the Discussions section on what happened as a result of these findings? Do the respective institutions have a concerted effort to making earlier IV-to-PO antimicrobial switches with this data?
MINOR COMMENTS:
- Syntax comments
- Line 49: “convulsively” – did authors mean “conversely”?
- Line 112: authors wrote “Initially” – but there was no follow-up as to what was done later? Maybe best to remove “initially” to avoid confusions.
- Line 128: Since only two of the authors collected demographics and clinical characteristics, how does this impact data analysis? It may be good to provide a clarification or to phrase that sentence differently.
Author Response
The authors have conducted a well-designed and analyzed study re: early oral antimicrobial switch for less critically ill adults with community-onset bacteremia in the ED setting. The study has a robust design and statistical analysis (i.e. a large sample size over the course of eight years, propensity score matching). It is a very well written paper - thank you for the contribution to this important infectious disease topic.
Response: Many thanks for your positive comments and substantial opinions.
MAJOR COMMENTS:
- Consider mentioning or referring to standardized term of “IV-to-PO” since this terminology is widely adopted in antimicrobial stewardship literature? Standardization of “IV-to-PO” may appeal more to readers due to familiarity.
Response: Thanks for your suggestion. The term of "IV-to-PO" was adapted throughout the revised manuscripts.
- Consider reviewing and including the following reference on IV-to-PO switch in bacteremia pts: Erickson, Reaghan et al. Open Forum Infect Dis. 2019 Nov 13;6(12):ofz490. doi: 10.1093/ofid/ofz490. eCollection 2019 Dec. Impact of an Antimicrobial Stewardship Bundle for Uncomplicated Gram-Negative Bacteremia.
Response: Thanks for your suggestion. The new reference was inserted as the Reference No. 7.
Were rapid diagnostics (i.e. MALDI-TOF, PCR, PNA-FISH, BC-GP) available for the study sites?
Response: Thanks for your question. During the study period, MALDI-TOP was available in one of the study hospitals. However, the period between blood culture sampling and pathogen identifications was 2 days. In the patients received early IV-to-PO switch, the median (IQR) of ED stay was 1.0 (0.3-2.2) days herein, as shown in Table 1. Therefore, the impacts of this rapid diagnostics on physician's decisions should be neglective.
Appreciate the cost analysis in Table 3, have authors consider direct costs (antimicrobial costs) vs. cost-effectiveness calculations (deaths averted, quality-adjusted life years gained (QALYs), incremental cost-effectiveness ratios (ICERs))?
Response: Thanks for your suggestion. However, focusing on short-term follow-up (30 days) after IV-to-PO switch, these cost-effectiveness calculations, in terms of QALYs and ICERs, were not appropriate to recognize the adverse impact of this switch. Therefore, antimicrobial cost, complication of antimicrobial therapy, and clinical therapeutic failure were regarded as our major outcomes. as shown in Table 3 (Page 14).
Also, consider calculations of defined daily dose (DDD) of antibiotics and/or days of therapy (DOT) since these are standard methods of quantifying antibiotic use? An extensive cost analysis could be a separate discussion or paper as a follow-up to the current manuscript.
Response: Thanks for your suggestion. In principal, the DDD was widely used for the comparisons of drug usage between different drugs in the specific patient populations or between different healthcare settings, or recognition the trend in drug utilization over time. Accordingly, DDD was inappropriately regarded as our outcomes. in our work, the day of therapy (DOT) in the early and late/no switch groups was listed in Table 1. For more reader's understanding, the daily cost of antimicrobial was inserted in the revised Table 3 (Page 14).
Early vs. late PO switch was not clearly defined. Some of the definition was covered in section 2.4
Response: Thanks for your opinions. For ED setting, the operation definition was adapted to define the early and late/no IV-to-PO switch. Based on ED physician's decision, patients remained on IV antimicrobial therapy after hospitalization through the ED were regarded as the late/no IV-to-PO switch group, whereas those that were switched to oral antibiotics from initial IV antimicrobial administration in the ED and then directly discharged from the ED and accepted regular outpatient clinic visits were categorized as the early IV-to-PO switch group. Please refer to line 126-131 on page 4.
Outcome Measures re: extremely short course vs. conventionally long-course. Consider clearly defining what is early vs. late switch in the context of this study. Duration of IV antimicrobial use before switch was not shared for the study population (i.e. median, range, etc.). There was some information in section 4. Discussion where the authors shared that extremely short course was approximately 1 day of antimicrobial IV administration. It would be helpful to include more information about this earlier in the paper.
Response: Thanks for your suggestion. As shown in Table 1, the median (IQR) duration of IV antimicrobial use before IV-to-PO switch in the early and late/no switch groups was 0.9 (0.3-2.0) and 6 (6-16), respectively. Accordingly, the switch from extremely short course (approximately 1 day) of appropriate antimicrobial IV administration was reasonably emphasized herein (Line 398-401, Page 16 and Line 469-701, Page 17-18).
Did each participating study site have an Antimicrobial Stewardship Program in place? Did the antimicrobial steward or pharmacists make the recommendations for the IV-to-PO switches or did the house staff make the switches on their own? It would be helpful to elaborate a bit about the resources and processes in place that allowed for the IV-to-PO switch to happen in the case other institutions would like to implement something similar.
Response: Thanks for your opinions and questions. During the study period, the Antimicrobial Stewardship Program was not performed in the ED of each participating hospital. So, to solve the problem of the ED overcrowding, we were interested in understanding the patient safety after extremely early IV-to-PO switch using the retrospective cohort. Based our finding, the further prospective study will be conducted after incorporation of an early switch strategy into the antimicrobial stewardship programme for bacteraemic patients diagnosed in the ED.
Would be interested in authors sharing in the Discussions section on what happened as a result of these findings? Do the respective institutions have a concerted effort to making earlier IV-to-PO antimicrobial switches with this data?
Response: Thanks for your opinions and question. As the above statements, we expect our finding will be contributory in reducing the ED overcrowding. Some sentences were inserted for our further plan (Line 461-464, Page 17 and Line 471-473, Page 18).
MINOR COMMENTS:
Syntax comments
- Line 49: “convulsively” – did authors mean “conversely”?
Response: The typo was corrected. To achieve more understanding, "in conclusion" was inserted (line 50, page 2).
- Line 112: authors wrote “Initially” – but there was no follow-up as to what was done later? Maybe best to remove “initially” to avoid confusions.
Response: Thanks for your question. This word was deleted (line 117-118, page4).
- Line 128: Since only two of the authors collected demographics and clinical characteristics, how does this impact data analysis? It may be good to provide a clarification or to phrase that sentence differently.
Response: Thanks for your question. For each patient, two authors inspected the medical records together for the consensus; and any discrepancies between authors in capturing medical information were resolved through discussion with another author. Please refer to line 143-146 on page 5.
Reviewer 3 Report
Overall, robust analysis of an interesting dataset looking at early antibiotic switching. Authors highlight the weaknesses and reach a fair conclusion.
Abstract
Line 49: Convulsively: think it is a typo, please replace.
Important to highlight these findings were amongst less critically ill patients Suggest mention this when describe matching criteria as not to mislead readers.
Introduction
Line 61: grammar: strategy
Methods:
Was ethical approval for retrospective analysis sought? Please add a sentence to justify decision.
Discussion:
Suggest strengths and weaknesses section on page 15/25 Line 378-397 be moved to discussion section where expected to find it
Author Response
Overall, robust analysis of an interesting dataset looking at early antibiotic switching. Authors highlight the weaknesses and reach a fair conclusion.
Response: Greatly thanks for your positive comments and substantial opinions
- Abstract
Line 49: Convulsively: think it is a typo, please replace.
Response: To achieve more understanding, "in conclusion" was inserted (line 50, page 2).
Important to highlight these findings were amongst less critically ill patients Suggest mention this when describe matching criteria as not to mislead readers.
Response: Thanks for your opinion. In the title (Page 1) and the section of abstract (Line 50, Page 2) and conclusion (Line 467, Page 17), the aimed population of less critically ill patents was emphasized for the reasonable adaption of the early switch strategy.
- Introduction
Line 61: grammar: strategy
Response: Thanks. The word was reworded (Line 63, Page 2).
- Methods:
Was ethical approval for retrospective analysis sought? Please add a sentence to justify decision.
Response: The sentence of ethical approval was inserted (Line 107-111, Page 4).
- Discussion:
Suggest strengths and weaknesses section on page 15/25 Line 378-397 be moved to discussion section where expected to find it
Response: This paragraph was moved to the section of discussion. Please refer to line 442-464 on page 17.
Round 2
Reviewer 1 Report
The responses to the comments, and the changes made, are adrquate.